# Strengthening suicide prevention: Evaluating a capacity building programme for community health workers in India

Soumitra Pathare[1] , Nikhil Jain[1], Deepa Pandit[1], Isha Lohumi[1] , Laura Shields-Zeeman[2,3] and Lakshmi Vijayakumar[4,5]

[1]Centre for Mental Health Law and Policy, Indian Law Society, India; [2]Department of Mental Health Prevention, Trimbos Institute, the Netherlands; [3]Faculty of Interdisciplinary Social Sciences, Utrecht University, the Netherlands; [4]Sneha Foundation Trust, India and [5]The Voluntary Health Services, India

## Research Article

**Keywords:**
suicide prevention; community mental health; community health workers; health system; gatekeeper training

**Corresponding author:**
Soumitra Pathare;
Email: spathare@cmhlp.org

## Abstract

The study evaluated a training programme adapted from the WHO mhGAP to enhance suicide prevention knowledge, attitudes, and confidence among 436 Community Health Workers (CHWs) in India. A pre–post intervention design assessed outcomes at four time points using a structured questionnaire, analysed via repeated-measures ANOVA. Mean knowledge scores increased from $6.32 \pm 0.14$ at baseline to $11.12 \pm 0.12$ post-training, then levelled off at $10.10 \pm 0.14$ and $10.10 \pm 0.13$ at 6 and 12 months, respectively; similarly, mean confidence scores increased from $4.96 \pm 0.11$ to $7.84 \pm 0.11$, remaining at $7.28 \pm 0.10$ and $7.44 \pm 0.10$ at the same time points. Mean attitude scores changed slightly from $41.00 \pm 0.38$ to $42.72 \pm 0.43$ over 12 months, indicating increased negative attitudes. Knowledge and confidence scores across time points were statistically significant ($p < 0.05$); however, this was not observed for attitude scores. Improvements were associated with CHW cadre and educational status. Post-training, CHWs demonstrated sustained improvements in knowledge and confidence for identifying, referring, and managing suicidal behaviour over 12 months, with those having lower baseline scores improving uniformly following the programme.

## Impact statement

Suicide is a major public health concern in India, especially in rural areas with limited access to care. Community health workers (CHWs), often the first point of contact in the public health system, typically lack suicide prevention training. While gatekeeper training has shown promise in improving knowledge and attitudes, evidence from low-resource Indian settings, particularly within public health systems, remains limited. This study shows that suicide prevention training adapted from the WHO mhGAP module led to sustained improvements in CHWs' knowledge, confidence and referral practices over 12 months in rural India. Unlike previous studies where effects declined, ongoing mentoring appeared to reinforce learning and practice. While attitudes remained largely unchanged, increased referrals suggest that CHWs were more engaged in identifying and supporting those at risk. These findings support integrating suicide prevention into CHW training under public health programmes. Sustained gains highlight the value of continued support over formal refreshers in low-resource settings. Future interventions should explore targeted strategies to shift attitudes and strengthen long-term behavioural change in community-based mental health care.

## Introduction

Suicide is a major public health crisis worldwide, with approximately 75% of suicides occurring in low- and middle-income countries (LMICs) (Vijayakumar and Phillips, 2016; Dandona et al., 2017). India has experienced a steady and significant rise in suicide rates over the past 5 years (Arya et al., 2022; Ganguli et al., 2025; National Crime Records Bureau, 2022), wherein the suicide has increased by 27.07% from 2018 to 2022 (Shanthosh et al., 2021; Kalne et al., 2022). These high and sustained figures underscore the need for more effective suicide prevention strategies suitable to the Indian context (Armstrong et al., 2011), delivered in different settings by a variety of service providers and stakeholders.

Community health workers (CHW) play a crucial role in delivering primary healthcare services in India (Armstrong et al., 2011; Shanthosh et al., 2021; Kalne et al., 2022). Their involvement has been effective in improving healthcare access, enhancing patient engagement, supporting screening and assessment at the community level and facilitating referrals for various physical health conditions (Shanthosh et al., 2021; Kalne et al., 2022). Research has also demonstrated that training

community healthcare workers to provide mental healthcare within their local contexts is an effective strategy (Armstrong et al., 2011; Shidhaye et al., 2019; Patel et al., 2021). India's National Suicide Prevention Strategy (2022) also emphasises the need to engage front-line community health workers in comprehensive suicide prevention efforts (Ministry of Health and Family Welfare GoI, 2021), thereby strengthening the public healthcare system's capacity to prevent suicides and attempted suicides.

Gatekeeper training (GKT) is a promising suicide prevention approach that has been recommended and adopted globally for community-based prevention strategies (Capp et al., 2001; Mann et al., 2005). While studies have explored GKT with various healthcare professionals, there is limited evidence on the implementation of such training for community health workers in India (da Silva Cais et al., 2011; Solin et al., 2021; Cross et al., 2022). The Suicide Prevention and Implementation Research Initiative (SPIRIT) employs this approach as a part of an integrated intervention package by training community healthcare providers in recognising, supporting, appropriately referring and following up with individuals identified as being at high risk for suicide (Pathare et al., 2020).

This paper reports on the impact of a suicide prevention training programme adapted from WHO's mhGAP Intervention Guide 2.0 (World Health Organisation, 2019) suicide module on the knowledge, attitudes, confidence and consequently practices of community health workers (Pathare et al., 2020). The study further examines variations in training impact across CHWs with differing professional backgrounds, aiming to provide insights into how contextual factors influence the outcomes of such capacity-building interventions in resource-limited settings.

## Methods

### Design

This research is part of a larger study, SPIRIT (Pathare et al., 2020), a cluster-randomised controlled trial designed to assess the impact of integrated suicide prevention interventions on the incidence of suicide and attempted suicide in intervention study sites as compared to control study sites in the rural district of Mehsana in Gujarat, India. SPIRIT consists of three interventions, namely (1) a secondary-school-based intervention to reduce suicidal ideation among adolescents, (2) a community storage facility intervention to reduce access to pesticides and (3) training for community health workers in recognition, providing appropriate support and appropriate referral and follow-up of people identified with high suicidal risk (Pathare et al., 2020). All three interventions were delivered to the village cohorts in the intervention arm, while in the control arm, only outcome measure data were collected.

### Intervention – suicide prevention training programme

The WHO mhGAP (World Health Organization Mental Health Gap Action Programme) intervention guide version 2.0 (mhGAP-IG 2.0) is an evidence-based mental health treatment guide for non-specialised health workers, primarily designed for low- and middle-income countries (LMICs) (World Health Organisation, 2019). Since mhGAP-IG 2.0 was not specifically designed for community health workers at the grassroots level, the research team undertook comprehensive adaptations to make it suitable for their competency levels and working environments.

The research team reviewed and contextualised the mhGAP-IG 2.0 to align with existing mental health care services and the skill levels of community health workers. Beyond language changes, the adaptation included cultural modifications to align with social constructs and understanding of suicide in India, as well as the addition of structured follow-up protocols for identified individuals and linked processes, which were not explicitly mentioned in the original mhGAP-IG 2.0. The team conducted two consultations with local stakeholders, including the District Public Health department, non-governmental organisations (NGOs), CHWs and people with lived experience, to assess training needs, psychosocial environment, prevalence of mental health problems and referral systems. After piloting the adapted material with CHWs in a neighbouring district, feedback led to streamlined content and the incorporation of role-play activities in the training programme.

The module and accompanying algorithms for self-harm and suicide detail the identification of high-risk behaviour, psychosocial support provision and referral pathways for individuals at risk of suicide.

### Training of trainers

The research team conducted a four-day Training of Trainers (ToT) to equip the trainers with the requisite confidence to facilitate and implement the training programme. The identification of trainers followed a systematic process. The research team engaged with various CHWs posted only across intervention sites. This was to ensure minimal interactions with CHWs from control sites and to prevent contamination across the study arms. A pool of 95 potential CHW trainers were approached and interviewed by Master Trainers, assessing their motivation, knowledge, rural community experience and availability. Fifty CHWs were shortlisted who were invited to attend a four-day training, conducted across two batches between July 2019 and August 2019.

The ToT programme comprised a blend of classroom instruction, practice in delivering the modules of the suicide prevention training programme, conducting role-plays and other interactive activities. This approach aimed to equip the selected CHW trainers with the necessary knowledge and confidence to effectively train other CHWs. It also aimed to prepare them to deal with difficult situations during the training, facilitation skills and de-stigmatisation of various aspects of suicide in a non-judgmental and supportive manner. This cascade approach allowed for efficient scaling of training to a large number of frontline providers, ensuring sustainability even beyond the SPIRIT project's lifespan. During the training, the master trainers evaluated participants across key domains, including engagement skills, conceptual understanding, knowledge retention, communication, motivation, teamwork and leadership. Fidelity and delivery quality were further assessed using a comprehensive checklist that captured all components of session delivery, from preparation and content adherence to participant engagement, responsiveness and session management. Based on performance across these assessments, 10 trainers were ultimately selected to deliver the intervention.

### Training curriculum and implementation aids for the suicide prevention training programme

The 2-day training programme consisted of 13 main modules. It aimed to equip CHWs with knowledge and confidence to identify and provide support to individuals at risk of suicide at a community level.

The training included modules on understanding suicide, covering epidemiology, risk and protective factors, common

myths and principles of safe communication. CHWs were taught how to conduct a structured suicide risk assessment and were introduced to evidence-based psychosocial support strategies, including psychoeducation, behavioural activation, stress-management techniques, problem-solving and strengthening social support networks. Additional modules focused on identifying and initiating referrals to mental health specialists and social welfare services, as well as a structured protocol for conducting regular follow-ups with at-risk individuals and their caregivers, which included reassessing risk, reviewing coping strategies and supporting ongoing help-seeking. The curriculum also included a dedicated session on self-care, managing emotional burden and recognising burnout in themselves and peers. All modules were carefully adapted for the Indian rural context, using locally relevant examples, simplified language and culturally sensitive approaches to ensure feasibility and acceptability for CHWs.

The training included interactive talks, small group discussions, role-plays and skills practice, along with theoretical sessions.

Furthermore, the team trained CHWs to use implementation aids like an offline reminder-based mobile application and a follow-up dial. The mobile application was designed by the research team and developed by a third-party private IT organisation based in India. The application, once downloaded by the trained CHW, provided an interface to add case details of individuals, recording key details such as identification, assessment dates. The application automatically scheduled the follow-up date and integrates them into CHW's in-built phone calendar and sends reminders to ensure timely follow-ups. Additionally, this application also contained various resource materials like training manuals, hand-outs on community-based psychosocial support resources, protocols for handling emergencies, follow-up and handling difficult situations, etc., for CHWs' reference. The application was completely offline and used local phone storage for storing the data (see Figure 1).

For CHWs who did not have a smartphone, we trained them to use a follow-up dial to determine the follow-up dates based on the baseline assessment date. The follow-up dial was designed by an internal research team and consisted of two overlapping, movable circular dials. The outer dial had all the days of the month, whereas the inner, smaller dial had the frequency of nine follow-ups from the baseline assessment date. CHWs would align the baseline date with any particular day of the month to establish the next dates for the nine follow-ups they had to conduct with the identified at-risk individuals. Both the application and the dial were created in vernacular language (Gujarati) (see Figure 2).

### Participants

In India, CHWs are categorised into distinct cadres, each with specific roles and educational qualifications. Accredited Social Health Activists (ASHAs) operate at the community level, serving as a link between households and health services (Ministry of Health & Family Welfare, 2005). They are typically local women with a minimum of an 8th-grade education. Anganwadi workers (AWWs), under the Integrated Child Development Services (ICDS), focus on nutrition and early childhood care and generally have a secondary education (10th grade). Auxiliary Nurse Midwives (ANMs), stationed at Health Sub-Centres, provide maternal and child healthcare as well as family planning services. Multipurpose Health Workers (MPHWs) are responsible for disease control, sanitation and surveillance. Both ANMs and MPHWs are required to complete a two-year health diploma after 12th grade (Ministry of Health & Family Welfare, n.d.; Ministry of Health & Family Welfare, n.d.). Community Health Officers (CHOs), deployed at Health and Wellness Centres, serve as mid-level healthcare providers, delivering comprehensive primary healthcare and managing non-communicable diseases. They hold a bachelor's degree in nursing (Kerry Scott and

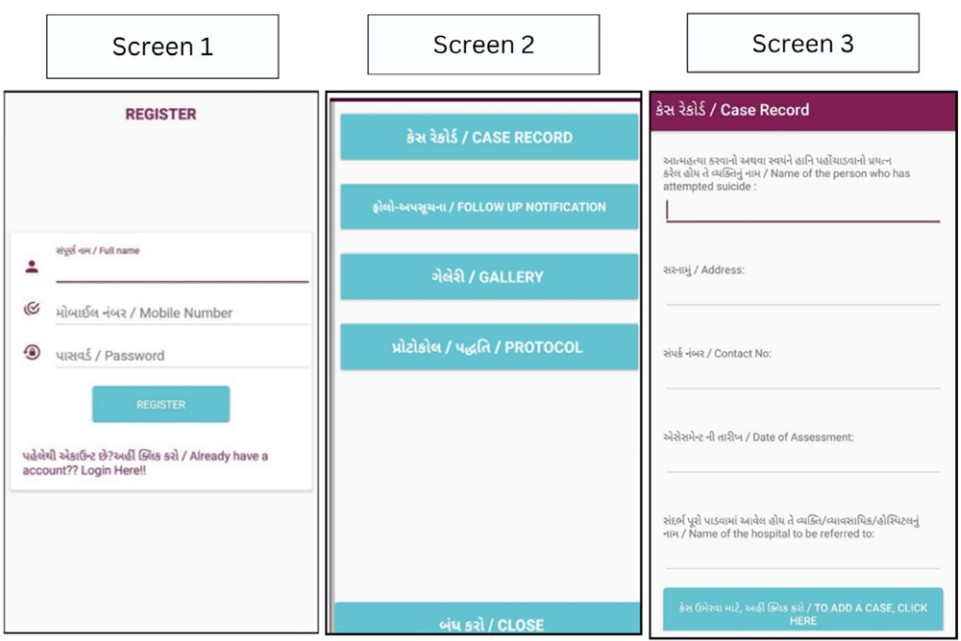

Screen 1 - Registration Interface, Screen 2- Modules Selection Interface, Screen 3 - Case Entry Interface

**Figure 1.** Offline Follow-Up Reminder Calendar Mobile Application Interface.

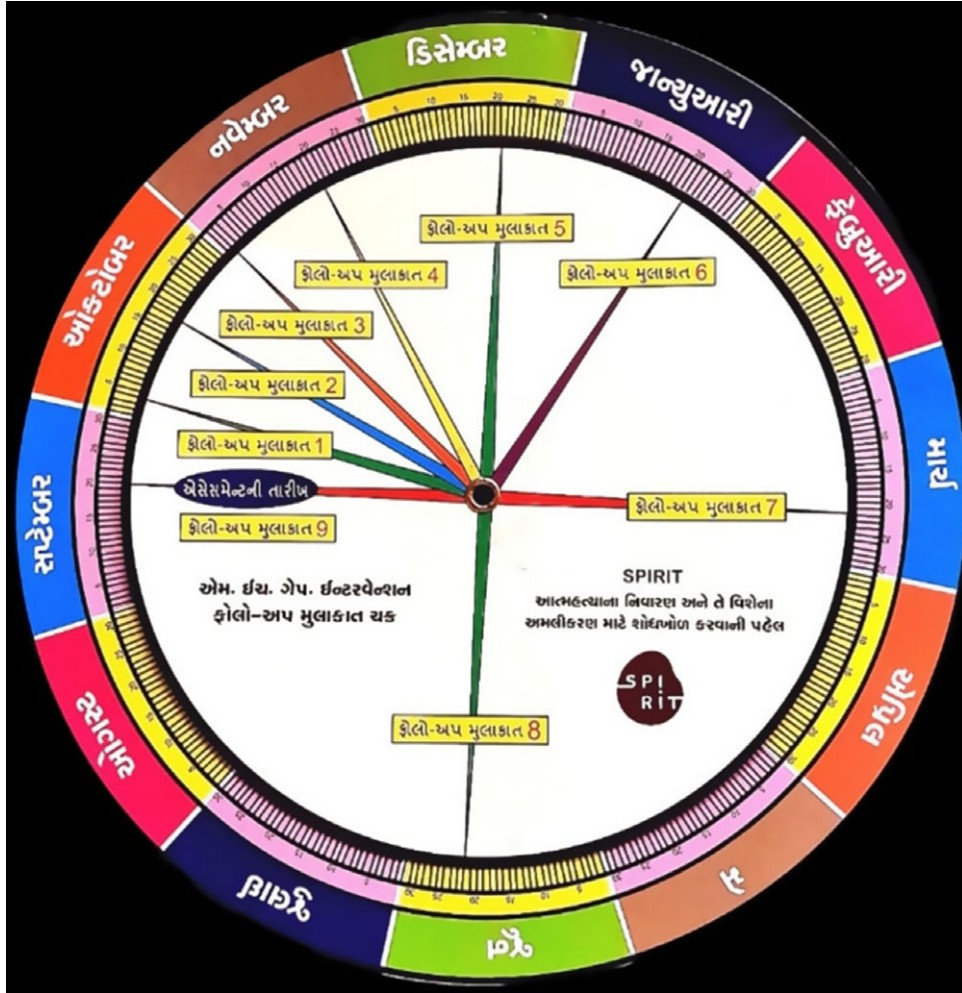

**Figure 2.** Follow-up dial.

Gergen, 2018; Ministry of Health & Family Welfare). Collectively, these cadres constituted the CHW workforce engaged in the SPIRIT intervention in Mehsana district.

The training for CHWs in the SPIRIT intervention was conducted in Gujarati by a team of 10 trainers under the direct supervision of the research team. A total of 20 batches participated in two-day training workshops over 22 months (November 2019 to September 2021), with 436 CHWs trained. Training participants included all CHW cadres. These CHWs, engaged in the SPIRIT trial, are employed by the Department of Health and Family Welfare and the Ministry of Women and Child Development, Gujarat, and were selected based on their current work postings across the study sites.

### Mentoring and support to CHWs

The CHWs were mentored and supervised by the research team in delivering the intervention. Monthly field visits were conducted by the field research team, and quarterly visits were made by researchers to monitor the implementation/use of the mhGAP adapted algorithm in practice. The field visits included hands-on supervision and case discussions, as well as regular assessments of protocol adherence and timely referrals. The project team provided guidance, support and feedback on CHWs' performance.

### Data collection and measures

The data collection was done using a self-reported structure tool in a group setting. The data were collected before and after the training programme (pre- and post-training) and took around 15–20 min to complete. Follow-up data were collected by researchers visiting the CHWs at their place of work 6 and 12 months from the training date (Refer to Annex 1).

The data collection tool was based on the questionnaire used by Armstrong et al. to examine the impact of training community health workers in mental health on knowledge, attitudes and skills (Armstrong et al., 2011). The knowledge section was assessed using a set of questions (n = 4) derived from previously validated questionnaires used in gatekeeper training surveys (Terpstra et al., 2018). The other section focuses on the confidence and abilities (n = 3) of CHWs to utilise their knowledge to identify and support individuals with a high risk of suicide in a timely and appropriate manner. Each item was scored on a 3-point Likert scale with responses – "don't know," "know a little," "have knowledge." The cumulative knowledge score ranges from 4 to 12, with higher scores indicating greater knowledge of the subject matter. The aggregate confidence score ranges from 3 to 9, with higher scores indicating a greater confidence in the application of their knowledge while working with at-risk community individuals.

Attitudes towards suicide prevention were assessed using a 14-item self-report questionnaire developed in the United States for use with frontline health care staff (Herron et al., 2016). The items are scored on a 5-point Likert scale (ranging from 1 to 5) reflecting the degree of agreement for all items with responses – "strongly disagree," "disagree," "uncertain," "agree," and "strongly agree." The questionnaire asked for the need for general mental health training rather than suicide prevention, as formative findings indicated that CHWs had not received any prior training in mental health. The total score ranges from 14 to 70, with an overall higher score indicating a more negative attitude towards suicide prevention.

The questionnaire was translated from English into the local language (Gujarati) and again back-translated to check for semantics. Thereafter, the questionnaire was tested as a pilot in smaller groups to ensure ease of understanding, feasibility and acceptability of the scale among the CHWs. Test–retest validity was employed to check the consistency and reproducibility of results obtained from the CHWs over time with a batch of 30 participants. It was observed to be high for a 14-item questionnaire with a correlation coefficient of 0.7 ($p < 0.001$) and Cronbach's alpha of 0.72.

### Data analysis

Data were analysed using the STATA 14.2 version, employing descriptive statistics (StataCorp, 2015). Frequencies with percentages were computed for categorical variables, while mean, standard deviation, median and interquartile range were calculated for continuous variables, based on the normality of the study variables. The data analysis employed a General Linear Model (GLM) for repeated-measures ANOVA, to examine changes within participants over time (baseline, post-training, 6 months and 12 months) and to compare how different groups performed over time after receiving the intervention. The assumptions underlying repeated-measures ANOVA were carefully examined prior to analysis. Normality of residuals for each outcome measure was assessed using Q–Q plots, which demonstrated an acceptable distributional fit, supporting the use of parametric methods. This approach allowed us to explore how scores evolved across time points within subjects and to examine differences between subjects, such as CHW categories, education levels and prior mental health experience.

### Findings

#### Socio-demographic characteristics of community health workers

Table 1 presents baseline differences in demographic characteristics for the community health workers. For analysis, we combined the three categories of ANM, MPHW and CHO into one as a "combined category," as there were small numbers in each category separately. CHWs from 54 study intervention sites attended and completed the baseline training (n = 436) out of a total of 450 CHWs posted in these sites. Around 14 (4.12%) CHWs could not attend any batches of training during the roll-out phase as they were on extended leave (sick or maternity leave). On average, the CHWs had a median of 12 years of education and 11 years (11.1 ± 8.2) of work experience in their community. Eighty-nine per cent of CHWs had not received any formal mental health training before the study, though 71.3% of CHWs expressed a need for mental health training as well.

### Impacts of training at pre- and post-training period and 6 months and 12 months

The impact of suicide prevention training on knowledge, attitude and confidence scores of CHWs was evaluated by comparing pre-training scores with post-training, 6-month and 12-month follow-up scores. The reduced response at 6 months (50% response rate) was largely due to COVID-19-related field access restrictions. Mean knowledge scores increased immediately post-training, rising from a baseline of 6.32 ± 0.14 to 11.12 ± 0.12. Although there was a slight decline over the subsequent months, the scores remained higher than the pre-training level, indicating a sustained impact till the study period. Similarly, mean confidence scores improved significantly from 4.96 ± 0.11 to 7.84 ± 0.11 post-training and remained consistent through the 6 and 12-month follow-ups, suggesting a sustained increase in confidence among participants in comparison to the pre-training level. Score variations for both of these aspects were statistically significant. In contrast, the mean attitude scores showed minimal variation throughout the study period. These details are mentioned in Table 2. Additionally, a limited number of individuals were supported before the training was imparted (n = 16), compared to 108 individuals supported after the training; this increase could be interpreted as an impact of the training (Vijayakumar et al., 2025).

### Variation of training effectiveness

The combined category group (comprising ANMs, MPHWs and CHOs, who all receive more education and training) scored significantly higher on knowledge and confidence than ASHA workers. However, no significant differences were observed between ASHA workers and AWWs or between AWWs and the combined category. Similarly, the combined category demonstrated a significantly better attitude towards suicide prevention compared to ASHA workers, indicating a more positive and proactive stance on this issue within the combined category group. Participants with higher education (≥15 years) scored significantly better on knowledge compared to those with lower education (<10 years), highlighting the positive impact of education on knowledge and confidence levels. Attitude scores were also notably better in the higher education group, indicating a more favourable attitude associated with increased educational attainment. Prior mental health training, which was operationalised as a binary self-reported measure indicating whether a CHW had ever participated in any mental health–related training prior to the SPIRIT intervention, was examined in relation to knowledge and skills outcomes. Although differences in scores by prior training status were not statistically significant, CHWs who reported having received any previous mental health training demonstrated higher mean knowledge scores compared to those without such training.

A repeated-measures ANOVA showed a significant interaction between time and education level for ATSP score, indicating that changes over time varied significantly across educational groups, with higher education groups (>15 years) associated with more positive attitudes (F = 1.90; $p = 0.04$; $n^2p = 0.02$). Similarly, there was a significant interaction between time and prior mental health experience for knowledge score, suggesting that changes over time differed significantly across these groups (F = 2.8; $p = 0.03$; $n^2p = 0.04$). For all other interaction effects, the results suggest that score changes over time followed a similar pattern across groups, with no single group exhibiting a unique pattern of improvement or decline (Table 3.).

**Table 1.** Baseline socio-demographic characteristics (n = 436)

| Variables | ASHA (n = 156) | AWW (n = 156) | Combined category (n = 124) | | | Total (n = 436) |
| | | | ANM (n = 26) | MPHW (n = 76) | CHO (n = 22) | |
|---|---|---|---|---|---|---|
| **Age, years** | | | | | | |
| Mean ± SD | 37.5 ± 8.2 | 44.9 ± 8.7 | 38.2 ± 10.2 | 36.5 ± 8.4 | 28.4 ± 4.2 | 39.6 ± 9.5 |
| Median (Q1, Q3) | 36 (31. 44) | 47 (38, 52) | 36 (31, 45) | 34 (31, 40) | 27 (26, 30) | 38 (32, 48) |
| **Gender** | | | | | | |
| Male | 0 | 0 | 0 | 43 (56.6) | 3 (13.6) | 46 (10.6%) |
| Female | 156 (100%) | 156 (100%) | 26 (100%) | 33 (43.4) | 19 (86.4) | 390 (89.4%) |
| **Years of education*** | | | | | | |
| Mean ± SD | 10.4 ± 2.6 | 12.4 ± 3.1 | 15.0 ± 2.4 | 14.1 ± 3.2 | 15.2 ± 2.9 | 12.3 ± 3.3 |
| Median (Q1, Q3) | 10 (9, 12) | 12 (10, 15) | 15 (14, 17) | 15 (13, 17) | 16 (15, 16) | 12 (10, 15) |
| **Years of education*** | | | | | | |
| <10 years | 42 (27.1) | 19 (12.3) | 0 | 4 (5.4) | 1 (4.5) | 66 (15.3) |
| 10–11 | 63 (40.6) | 40 (25.8) | 1 (3.8) | 7 (9.5) | 0 | 111 (25.7) |
| 12–14 | 36 (23.2) | 47 (30.3) | 12 (46.2) | 19 (25.7) | 1 (4.5) | 115 (26.6) |
| > = 15 | 14 (9.0) | 49 (31.5) | 13 (50.0) | 44 (59.5) | 20 (90.9) | 140 (32.4) |
| **Work experience (years)** | | | | | | |
| Mean ± SD | 7.2 ± 4.8 | 16.7 ± 8.4 | 10.9 ± 9.8 | 9.8 ± 6.7 | 3.2 ± 2.6 | 11.1 ± 8.2 |
| Median (Q1, Q3) | 7 (3, 10) | 18 (10, 24) | 9 (4, 13) | 8 (6, 11) | 3(1, 5) | 9 (5, 15) |
| **Work experience (years)** | | | | | | |
| <5 | 51 (32.7) | 13 (8.3) | 7 (26.9) | 8 (10.5) | 15 (68.2) | 94 (21.6) |
| 5–9 | 50 (32.1) | 24 (15.5) | 10 (38.5) | 43 (56.6) | 6 (27.3) | 133 (30.5) |
| 10–14 | 45 (28.8) | 23 (14.7) | 3 (11.5) | 16 (21.1) | 1 (4.5) | 88 (20.2) |
| > = 15 | 10 (6.4) | 96 (61.5) | 6 (23.1) | 9 (11.8) | 0 | 121 (27.8) |
| **Formal training in mental health** | | | | | | |
| Yes | 26 (16.8%) | 10 (6.4%) | 4 (16.0) | 4 (5.3) | 0 | 45 (10.4%) |
| No | 129 (83.2%) | 146 (93.6%) | 21 (84.0) | 72 (94.7) | 22 (100%) | 389 (89.6%) |
| **Need for training in mental health** | | | | | | |
| Very much needed | 108 (69.2%) | 108 (69.2%) | 19 (73.1) | 56 (73.7) | 19 (86.4) | 310 (71.3%) |
| Somewhat needed | 42 (26.9%) | 34 (21.8%) | 6 (23.1) | 17 (22.4) | 3 (13.6) | 102 (23.4%) |
| Not needed | 5 (3.2%) | 13 (8.3%) | 0 | 2 (2.6) | 0 | 20 (4.6%) |
| Not sure | 1 (0.6) | 1 (0.6) | 1 (3.8) | 1 (1.3) | 0 | 4 (0.8%) |

Values are presented as mean ± SD as well as median interquartile range. ASHA: Accredited Social Health Activist; AWW: Anganwadi Workers; Combined category: Auxiliary Nurse Midwife (n = 26) and Multipurpose Health Workers (n = 76), and Community Health Officers (n = 22).
*Data available only for 432 CHWs on the "years of education" variable.

## Discussion

The primary objective of this study was to investigate the impact of suicide prevention training on knowledge, attitudes and practices among CHWs in rural India (Pathare et al., 2020). The training led to improvement in mean knowledge and confidence scores immediately after the training, which are consistent with earlier studies that indicate training can effectively enhance the mental health literacy of non-specialist healthcare professionals in LMICs (Armstrong et al., 2011; Gureje et al., 2015; Caulfield et al., 2019). The mean scores sustained during the follow-up time points (6 months and 12 months) and remained significantly higher than pre-training levels, indicating a sustained impact for the study period. These findings differ from other gatekeeper training programmes, where

follow-up evaluations typically conducted at a third assessment time point (6th or 8th month) have indicated a decline in the training's effect over time (Bell, 2015; Reiff et al., 2019; Holmes et al., 2021). Previous studies have suggested that incorporating a refresher session around the 6-month mark can help sustain improvements in knowledge and competency over time for any training programme (Holmes et al., 2021). In the current study, a similar effect may have been achieved through ongoing support and mentoring by the research team, which likely helped maintain CHWs' confidence reducing the need for a formal refresher session.

Conversely, attitude scores about suicide prevention were largely unchanged over the study period. Although this suggests that no significant change has occurred, it may reflect the deep-seated nature

**Table 2.** Comparison of scores for CHWs (n = 436) across different time points

| | Pre-training (n = 436) | Post-training (n = 436) | At the end of 6 M (n = 217) | At the end of 12 M (n = 369) |
|---|---|---|---|---|
| Knowledge score total | 6.3 ± 0.1 | 11.1 ± 0.1[a] | 10.1 ± 0.1[b] | 10.1 ± 0.1[c] |
| ASHA | 6.0 ± 0.2 | 10.5 ± 0.2[a] | 9.9 ± 0.2[b] | 10.0 ± 0.2[c] |
| AWW | 6.3 ± 0.2 | 11.3 ± 0.1[a] | 9.8 0.2[b] | 10.0 ± 0.2[c] |
| Combined category | 6.7 ± 0.3 | 11.3 ± 0.2[a] | 10.6 ± 0.3[b] | 10.3 ± 0.3[c] |
| Confidence score total | 4.9 ± 0.1 | 7.8 ± 0.1[a] | 7.2 ± 0.1[b] | 7.4 ± 0.1[c] |
| ASHA | 5.1 ± 0.2 | 7.5 ± 0.2[a] | 7.3 ± 0.2[b] | 7.0 ± 0.2[c] |
| AWW | 4.7 ± 0.2 | 8.0 ± 0.2[a] | 7.1 ± 0.1[b] | 7.7 ± 0.1[c] |
| Combined category | 5.2 ± 0.2 | 7.9 ± 0.2[a] | 7.6 ± 0.2[b] | 7.6 ± 0.2[c] |
| Attitude towards Suicide prevention (ATSP) score total | 41.0 ± 0.3 | 41.3 ± 0.5 | 42.2 ± 0.4 | 42.7 ± 0.4 |
| ASHA | 41.3 ± 0.6 | 41.4 ± 1.0 | 43.2 ± 0.8 | 43.7 ± 0.7 |
| AWW | 41.3 ± 0.5 | 42.3 ± 0.8 | 42.7 ± 0.6 | 42.9 ± 0.6 |
| Combined category | 39.8 ± 0.8 | 39.3 ± 1.2 | 39.9 ± 0.9 | 40.8 ± 0.9 |

Values are presented as mean ± SE.
[a]Comparison with baseline, $p < 0.05$.
[b]Comparison between baseline and 6 M, $p < 0.05$.
[c]Comparison between baseline and 12 M, $p < 0.05$, within subjects' comparison overtime not significant, $p = 0.80$.

**Table 3.** Comparison of scores across different CHW, educational and prior mental health training categories

| | | Knowledge score | Confidence score | Attitude score |
|---|---|---|---|---|
| CHW categories | ASHA | 9.14 ± 0.13 | 6.73 ± 0.11 | 42.41 ± 0.51 |
| | AWW | 9.40 ± 0.11 | 6.89 ± 0.09 | 42.33 ± 0.43 |
| | Combined category | 9.76 ± 0.15[a] | 7.08 ± 0.13[a] | 40.0 ± 0.61[a] |
| Educational categories | < 10 years of education | 9.20 ± 0.17 | 6.89 ± 0.14 | 41.75 ± 0.67 |
| | 10–11 years of education | 9.24 ± 0.14 | 6.67 ± 0.12 | 42.83 ± 0.57 |
| | 12–14 years of education | 9.30 ± 0.15 | 7.10 ± 0.13 [x] | 42.66 ± 0.61 |
| | > = 15 years of education | 9.75 ± 0.13[x] | 6.94 ± 0.11 | 40.53 ± 0.52 [x] |
| Prior mental health training | Received training | 9.68 ± 0.26 | 6.92 ± 0.21 | 41.26 ± 1.02 |
| | Has not received training | 9.38 ± 0.08 | 6.87 ± 0.06 | 41.88 ± 0.31 |
| | *p* value | *0.26* | *0.83* | *0.56* |

Values are presented as mean ± SE, a = comparison between ASHA and combined category $p < 0.05$.
ASHA, accredited social health activist; AWW, anganwadi workers; Combined category: Auxiliary Nurse Midwife, Multipurpose Health Workers and Community Health Officers; x = comparison with < 10 years of education $p < 0.05$.

of attitudes, particularly for complex and sensitive topics such as suicide which often require more in-depth engagement and sustained reinforcement than can be achieved through brief training interventions (Patel et al., 2018). Notably, there was an increased referral to specialised mental health care and helplines at the 6-month and 12-month marks following the training. This increase suggests that despite minor attitude shifts, CHWs became more engaged in identifying and referring individuals at risk of suicide, reinforcing the effectiveness of the training in enhancing practical intervention skills.

### Influence of education, CHW cadres and prior training exposure

CHWs with higher educational backgrounds (ANMs, MPHWs, CHOs) demonstrated greater improvements in knowledge, confidence and attitude compared to ASHA and AWWs. This suggests that these higher-cadre CHWs are more likely to have had prior exposure to training programmes as part of their professional roles, which may have contributed to their ability to engage with complex health topics (Ministry of Health & Family Welfare; Ministry of Health & Family Welfare). Prior research indicates that CHWs with greater educational backgrounds tend to have a higher capacity for mental health service integration (Saxena et al., 2007; Patel et al., 2018). This is particularly relevant for the CHOs, ANMs and MPHWs, who, due to their clinical responsibilities, may have had prior interactions with mental health cases, making them more receptive to new knowledge and confident to identify, refer and manage suicidal behaviour. In contrast, AWWs and ASHAs, who primarily focus on maternal and child health with little exposure to mental health training, may have faced greater challenges in acquiring and integrating suicide prevention knowledge. This is further supported by studies that indicate that frontline health workers with minimal exposure to mental health services often require more intensive training to effectively engage in such interventions (Singla et al., 2017; Kohrt et al., 2018).

These findings emphasise the importance of structured and differentiated training approaches for CHWs, ensuring that all cadres, regardless of their background, can effectively engage in suicide prevention efforts. Future training programmes must incorporate continuous support, refresher training and peer mentorship networks to facilitate long-term improvements in knowledge, confidence and attitudes. At the same time, CHW training for suicide prevention may represent an additional burden for CHWs who already manage extensive responsibilities with limited compensation, and expanding CHW roles without commensurate structural support, supervision and remuneration risks exacerbating workload strain and compromising long-term sustainability. In this study, the training was implemented in collaboration with local health systems and integrated into routine CHW roles, helping to minimise additional workload; however, careful planning remains essential to ensure feasibility and sustainability when scaling such interventions.

Targeted training strategies must also be formulated for CHWs with lower educational levels and no previous mental health exposure to fill the learning gap and maximise training benefits. Given the role of CHWs as the first point of contact for many at-risk individuals, strengthening their mental health literacy through context-specific training will be critical in resource-limited settings (World Health Organisation, 2021). Improving CHWs' knowledge and confidence is crucial for the early identification and prompt intervention, two essential elements of effective suicide prevention strategies.

## Study strengths and limitations

The research showed a high rate of participation in the training, indicating high levels of engagement from CHWs and their interest in improving their skills in suicide prevention. One of the strengths was that the measures were created in India, and the algorithm was systematically adapted to the Indian context through a rigorous process, making it culturally and contextually relevant. The research also had an extensive assessment approach to evaluating change in knowledge, attitudes and practices at the pre-, immediately post-training, and at 6 months and 12 months post-training time points, with important findings about both short- and long-term retention and effect. Using a relatively high number of samples, the study is more consistent and generalisable compared to existing literature.

However, the study had some limitations. The use of self-reported measures for measuring knowledge, attitudes and practices could have introduced response bias or social desirability bias. We conducted face validity checks and translation/adaptation of the scale to ensure its appropriateness for the study context. However, a more detailed validation (construct validity and internal consistency check) was not conducted within the scope of this study and is therefore acknowledged as a limitation. The lack of independent verification for referrals made by CHWs may have affected the accuracy of reported outcomes. Like most training-based interventions, variations in facilitator effectiveness, participant engagement and contextual variables could have impacted the overall effectiveness of the training. Additionally, skills were assessed through self-reported confidence and reported practice rather than direct observation, which limits conclusions about actual skill acquisition.

## Conclusion

The improvements in knowledge and confidence among CHWs support the feasibility and effectiveness of adapting the mhGAP module for suicide prevention training and implementing it through CHWs in rural Indian contexts. This training equips CHWs with the necessary tools and knowledge, enabling them to act as gatekeepers who can identify and refer individuals at high risk of suicide to appropriate services. Such a task-shifting approach addresses the gap in mental health service provision in resource-constrained settings, where access to specialist care is limited. As attitudes did not demonstrate statistically significant improvement over the follow-up period, future research should focus on evaluating long-term outcomes and identifying strategies to strengthen training approaches that can better address attitudinal dimensions, while sustaining improvements in knowledge and confidence.

**Open peer review.** To view the open peer review materials for this article, please visit http://doi.org/10.1017/gmh.2026.10188.

**Data availability statement.** De-identified data will be made available to researchers or anyone on request to one of the principal investigators following the data sharing policies of the institute.

**Author contribution.** S.P. had full access to all data in the trial and takes responsibility for the integrity of the data and the accuracy of the data analysis. S.P., N.J. and I.L. conceptualised the study. S.P., N.J., I.L. and D.P. contributed to data acquisition, analysis and interpretation. All authors drafted the manuscript. All authors contributed to the study implementation, reviewed drafts and were involved in interpretation and writing. Statistical analysis was conducted by D.P. and N.J. Administrative and technical support was provided by S.P., N.J. and I.L. Supervision was led by S.P., L.V. and L.S.Z.

**Financial support.** This research is funded by a US National Institute of Mental Health (NIMH) grant no. 5U19MH113174-03. The funder of the trial had no role in trial design, data collection, data analysis, data interpretation, or writing of the manuscript.

**Competing interests.** None.

**Ethics statement.** Informed consent was obtained from all participants who completed the pre- and post-training questionnaire. Ethical approval for this study was obtained from the Institutional Review Board (IRB) of the Indian Law Society (ILS/36/2017) on 22 March 2017.

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
