## [Reviewer Report]

This manuscript documents impact evaluation of a capacity-building program aimed at community health workers (CHWs) in India to address suicide prevention. It is part of a cluster randomised control trial (RCT). The manuscript could benefit from further details and analysis as detailed below.

1. The CHWs in the intervention arm of the RCT were trained as part of the intervention. The manuscript presents the “impact” of the capacity-building program based on before-and-after measures in the trained CHWs to determine if observed changes are due to the program. No comparative data from CHWs in the control arm of RCT are presented, which will be important to isolate the impact of the capacity-building program in the intervention arm. Also, there is no discussion on the potential confounding factors and how those were addressed in the analysis.

2. The primary paper on RCT (Pathare et al 2020) indicates process measures under this intervention. Those are not presented to situate the findings presented.

3. Self-reported measures are used, in particular for the people with suicide risk served, to document the change over time as a result of the capacity-building program. Given that this is RCT, one would expect the outcomes to be measured accurately and consistently across intervention and control groups. Self-reported outcomes can inflate the apparent impact if participants in the intervention arm report more optimistically. This can threaten internal validity.

4. The training program led to a significant improvement in CHWs’ knowledge demonstrating its effectiveness in knowledge transfer. However, there was no measurable change in their attitudes towards suicide prevention suggesting that additional interventions may be needed to influence beliefs and motivation. This finding needs to be explored and explained more as it will have implications on sustainability of the program and perhaps on the overarching aim of suicide prevention in the community. Also, attitude was assessed using a questionnaire developed in the Unites States. Please elaborate if this was validated for use in the current setting, and if so how.

5. Please provide more details on the training curriculum and tools (section 2.4 in the manuscript). The current content is generic. Lines 137-157 provide details on tools that the CHWs were trained in but no data from this are presented or used in the assessment tools.

6. Table 1 – please provide break-up of the combined category as it totals to 124. It would be more informative to provide the variables shown in the table by the cadre of CHW than overall as shown in the Table currently.

7. Table 2 – Please provide data by cadre of CHWs. Also, please comment on the drop-out in response rate for 6 months (50% response rate) and 12 months (85% response rate) time periods. 50% response rate does not allow for reasonable comparison. It would be useful to understand the participant demography at 6 months.

8. Table 3 – Please provide data by cadre of CHWs. How does one explain the findings under “prior mental health training”?

9. Table 4 does not add value to the analysis presented.

---

## [Reviewer Report]

Dear Authors, I appreciate the work that you have carried out. Please find the comments below :

Abstract - The statistical significance of the findings needs to be mentioned.

Intervention - were any guidelines followed for adaptation, not involving persons with lived experience during intervention design, can be a limitation. How was the quality & fidelity of the intervention delivery during the ToT exercise ensured?

Place - Mehsana (State’s name can also be mentioned here)

Skills- the questionnaire items largely cover knowledge & its application. From a learning-teaching perspective, it can’t be called skills (which is the psychomotor or practice aspect).

Analysis - the test-retest reliability can’t provide internal consistency (Cronbach’s alpha). Please clarify.

also mention about normality of data assessment -as for non-parametric data ANOVA can’t be applied

Discussion - referral to the specialist: through which variable was it measured in the current study?

Limitation - skill was not assessed through a validated means.

I hope these comments will help you improve your manuscript.

Regards, Reviewer

---

## [Reviewer Report]

This a pretty neat study. It does what it says and is written quite well. I only have a few queries but other then that, well done to the authors!

Introduction:

Lines 21-23: You should ideally be citing papers here that show an increase in suicide rates in India in recent years. For example: Arya V, Page A, Spittal MJ, Dandona R, Vijayakumar L, Munasinghe S, John A, Gunnell D, Pirkis J, Armstrong G. Suicide in India during the first year of the COVID-19 pandemic. Journal of affective disorders. 2022 Jun 15;307:215-20; Ganguli D, Singh P, Das A. Decriminalizing suicide: the 2017 Mental Healthcare Act and suicide mortality in India, 2001–2020. Cambridge Prisms: Global Mental Health. 2025 Jan;12:e74.

Design:

I know the term “suicide death” is commonly used these days but I do think it is a tautology as suicide, by definition, means death. It should just be suicide.

Training of trainer:

Line 108: “It also aimed to prepare them to deal with the difficult situation during the training”. Why “the” difficult situation? It is a typo, I am guessing

Please make sure the term “Master trainer” is used consistently as sometimes the letter “m” in “Master trainer” is uppercased and sometimes it’s not

Line 128: You refer to Mehsana district for the first time here and it kind of comes out of the blue as you haven’t disclosed the name of the district before. I suggest you mention it much earlier and let the readers know that this is where the study is based.

How was their time managed as this is something extra?

Data collection and measures:

Line 195: I suggest you describe exactly what it is that you are measuring among the participants before you describe the data collection process.

Lines 221-226: Hmmm, I am a bit sceptical about the “changes in CHW practice at 6- and 12-month follow-up” because the number of people at suicide risk identified can simply suggest that a CHW did not come across any person at suicide risk which would have nothing to do with the training they received and hence would be an incorrect representation of the impact of training. This could have been partially mitigated had there been some data prior to the training on the number of people CHWs identified at risk of suicide unless ofcourse, it would have been zero or close to it. I am guessing that is the case? Even so, this should at least be mentioned in the limitations section of the study.

Lines 242-243: This is a bit pedantic, but I would not say “as a result of the intervention exposure”. I would rather say something like “after receiving the intervention” or something along those lines. What you have written makes it feel too definitive and the only thing that can have an impact during the follow up period which is never strictly true.

Line 268: Please don’t say “indicating a lasting impact”, stick to 12 months. You don’t know what happens after 12 months hence cannot use the word ‘lasting’.

Line 288-289: you use ‘although’ twice in the same sentence, please correct.

Line 310-311: again, remove ‘lasting impact’

I was wondering if this can be deemed as an “extra burden” for CHWs on top of everything they already do for relatively low compensation. You should address this somewhere in the manuscript because as well-meaning as all of this is, it can be an added responsibility to those already burdened with enormous work.

---

## [Editor Report]

Dear authors,

Thank you for your submission to the special issue. We have obtained three reviews from experts in the field, and as you can see, while they are generally favourable, they have several queries. Please address them in detail, responding to each one. I look forward to reading your revised manuscript. 

Thank you and all the best,

Dr. Sandersan Onie

---

## [Reviewer Report]

Well done. For next time though, please attach a word file with each of the reviewer comments and your responses along with the revised manuscript. I had minor comments so it was OK to look at the revised manuscript but it would have been very difficult to do it if I had major/extensive comments.

---

## [Editor Report]

Dear Prof Pathare, 

Thank you for your extensive revisions. I have one final comment. In your conclusion, you state a ‘significant improvement’. As the term ‘significant’ can be subjective, please remove, or specify if it is a statistical significance that is meant.

I look forward to receiving your revised manuscript.

Thank you and all the best,

Dr. Sandersan Onie

---

## [Editor Report]

Dear Prof Pathare,

Thank you for your revised submission. I am now happy to recommend this manuscript for publication. Once again, thank you for your continued contribution to the field.

All the best,

Dr. Sandersan Onie